# The Response of the miRNA Profiles of the Thyroid Gland to the Artificial Photoperiod in Ovariectomized and Estradiol-Treated Ewes

**DOI:** 10.3390/ani15010011

**Published:** 2024-12-24

**Authors:** Zizhen Ren, Wei Wang, Xiaoyun He, Mingxing Chu

**Affiliations:** State Key Laboratory of Animal Biotech Breeding, Institute of Animal Science, Chinese Academy of Agricultural Sciences, Beijing 100193, China; rzz100805@163.com (Z.R.); wangw8182@163.com (W.W.); hexiaoyun@caas.cn (X.H.)

**Keywords:** sheep, thyroid gland, photoperiod, miRNA

## Abstract

Photoperiod is a key factor in sheep reproduction, and the mechanism of the thyroid gland in reproduction is unknown. In this study, RNA-seq and bioinformatics analysis were used to explore the effects of different photoperiods on thyroid miRNA expression. In total, 105 differentially expressed miRNAs were found, most of which were new miRNAs. Functional analysis showed that the photoperiod response was related to olfactory, Wnt, and Apelin signaling pathways. LncRNA-mRNA-miRNA network analysis reveals miRNA targets. This study provides new insights into the function of miRNA in sheep thyroid under the photoperiod.

## 1. Introduction

Sheep are short-day estrous animals, and most sheep breeds are in the estrus and reproduction stage in autumn and winter; seasonal estrus is a key factor limiting the reproductive performance of ewes. For example, Sunite showed obvious seasonal estrus. August to March of the same year and April to July of the same year were its estrus times [1]. Sheep with seasonal estrus can only have one birth in one year, which greatly affects the number of lambs and milk. In sheep, light stimulates the retina and transmits signals to the pineal gland through the suprachiasmatic nuclei (SCN), leading to changes in the duration of melatonin (MEL) secretion using the pineal gland during the night. Melatonin Receptor1 (MT1), expressed in pars tuberalis (PT), senses the change in MEL, and a series of light signal molecules in the PT region is changed, resulting in the secretion of the gonadotropin-releasing hormone (GnRH) in the hypothalamus to activate or close the hypothalamic–pituitary–gonadal (HPG) axis, thereby regulating the seasonal breeding behavior of sheep [2,3]. Fertility is a complex biological trait regulated by multiple organs and glands. In this process, thyroid hormones secreted by the thyroid gland directly act on the reproductive system of animals through specific nuclear receptors, such as ovaries, placenta, and uterus [4], and thus they have an important impact on the development of oocytes [5].

MicroRNAs (miRNAs) are a class of non-coding single-stranded RNA molecules with a length of about 22 nucleotides encoded by endogenous genes. They are known regulatory factors in the basic biological processes of animals and plants. By being paired with the 3′-UTR of the target gene, it promotes mRNA degradation or inhibits mRNA translation, thereby inhibiting the expression of its target gene. miRNAs are an important member of the thyroid transcriptome and also play an important role in reproduction. The reproductive regulation mechanism of miRNA in plateau animals [6], zebrafish [7], cattle [8], human [9], sheep [10], and other animals have been studied. The normal operation of the thyroid gland is very important for the development of the body [11,12,13,14], and miRNA is significantly related to some functions of the thyroid gland [15,16,17]. In addition to being a marker of disease, miRNA can also be used as a therapeutic tool for some diseases [18,19]; it has great potential in diagnostic and prognostic biomarkers. MicroRNAs in T-cell immunotherapy [20] act as a target for the treatment of cardiovascular diseases [21], and miR-297 can inhibit the tumor progression of liver cancer by targeting PTBP3 [18].

At present, researchers have little understanding of how the photoperiod stimulates the molecular neuroendocrine axis and the specific mechanisms of seasonal changes in reproductive behavior. Although the studies on species such as hamsters, cattle, and sheep mainly focused on changes in hormone levels, the expression patterns of key genes and proteins in long photoperiod and short photoperiod environments were also preliminarily compared [22,23]. However, there is still a lack of any comprehensive understanding of the effects of the photoperiod at the level of thyroid transcriptome. Since 1983, the study on the regulation of the photoperiod on seasonal estrus traits in sheep has been initiated [24]; so far, bilateral ovariectomy models have been widely used to explore the organ and tissue functions of mammals, such as rats, mice, goats, and sheep. Therefore, this study collected thyroid tissues of three groups of nine ewes to analyze the expression mechanism of potential thyroid miRNAs. Based on our previous research work, this study used high-throughput sequencing technology and bioinformatics analysis to explore the effect of photoperiod changes on thyroid transcriptome [25].

## 2. Materials and Methods

### 2.1. Ethics Statement

The experimental animals in this study complied with animal welfare standards. The research follows the relevant animal welfare guidelines and regulations to ensure that the pain of animals is minimized during the experiment.

### 2.2. Sample Collection

The ewes were treated with OVX + E2 and raised on a farm at the Institute of Animal Husbandry and Veterinary Medicine, Tianjin Academy of Agricultural Sciences, Tianjin, China [26]. In this study, we randomly selected 9 2–3-year-old, healthy, multiparous Sunit ewes. All ewes were placed under the same feeding conditions, ensuring that they could eat and drink freely. To explore the effect of the light cycle on the physiological state of ewes, we assigned 9 ewes to 3 different feeding groups (Table 1). The ewes in each room were marked as A1, A2, A3 (LP42), B1, B2, and B3 (SP42) and C1, C2, and C3 (SPLP42). On day 42, after slaughter, the thyroid gland was quickly removed from the brain, rinsed with PBS (pH 7.4), frozen in liquid nitrogen, and stored at −80 °C for subsequent transcriptome sequencing analysis (Table 2) and other subsequent studies.

### 2.3. RNA Extraction and Sequencing

Total RNA was extracted from experimental samples using TRIzol reagent (Invitrogen, Carlsbad, CA, USA). Then, the purity of RNA samples was detected using the Nano Photometer spectrophotometer (IMPLEN, Westlake Village, CA, USA). Finally, further analysis was performed using the electrophoresis and bioanalyzer 2100 system and RNA Nano 6000 Assay kit (Agilent Technologies, Santa Clara, CA, USA). RNA values greater than or equal to 7.5 can be used as the next result analysis. Subsequently, RNA was extracted for high-throughput sequencing. Then, the sequencing data were subjected to quality control, and Bowtie was used to compare it with the reference genome, and then the differentially expressed miRNAs were subjected to GO and KEGG pathway enrichment analysis; finally, Cytoscape was used to construct a co-expression network containing lncRNA, miRNA, and mRNA.

### 2.4. Data Quality Control and Sequence Alignment

To obtain high-quality data sets, we took a series of strict filtering steps. First, we excluded sequences that did not detect inserted fragments. Then, we removed the sequences whose poly-A tail length did not meet the standard (the proportion of continuous AT sequences reached 20% or the proportion of total AT sequences reached 80%). In addition, we also excluded sequences whose lengths were not within the preset requirements and those with lower quality. To further ensure the reliability of the data set, we also calculated the Q20, Q30 ratio, and GC content of the data set. All subsequent analysis and research will be based on this carefully screened clean data set. The reference genes and genome annotation files can be downloaded from the ENSEMBL website (http://www.ensembl.org/index.html, accessed on 15 October 2023), with Bowtie used to build the reference genome database, and then the clean data can be compared to the reference genome through Bowtie.

By comparing the clean reads in the sample with known reference sequences, and further comparing with the miRNA sequences of specific species in the miRBase database, the matching sequences of each region in the sample can be determined.

To identify unrecognized miRNAs, ncRNAs, and repetitive sequences, we used a method that annotates small RNA molecules derived from mRNAs by precisely matching clean reads with exons and introns of genes (requiring 100% overlap) [27].

### 2.5. GO and KEGG Pathway Enrichment Analysis of Differentially Expressed Genes

GO (Gene Ontology, http://geneontology.org/, accessed on 10 November 2023) functional enrichment analysis can determine the main biological functions of differentially expressed genes. KEGG (Kyoto Encyclopedia of Genes and Genomes, http://www.kegg.jp/, accessed on 17 November 2023) is a database of genome-wide and metabolic pathways. The hypergeometric test was applied to each pathway in KEGG for enrichment analysis, and the obtained *p* value was corrected using multiple tests. The threshold was *q* < 0.05, which was defined as a significant threshold. GO and KEGG databases were used to further study differentially expressed genes to study the function of genes and determine the pathways that they participate. If the *p* ≤ 0.05, the enrichment was considered significant.

### 2.6. Construction of Integral miRNA–mRNA Interaction Network

To predict the function of expressed genes in sheep reproduction, a network based on miRNA and mRNAs was established using Cytoscape (V3.8.2) [28].

### 2.7. Statistical Analysis

SPSS 25.0 statistical software was used to evaluate the experimental results via a *t*-test. All data are presented as mean ± standard error (SE). One-way analysis of variance was used for qPCR verification. *p* < 0.05 was considered statistically significant.

## 3. Results

### 3.1. Reference Sequence Alignment Analysis

Based on the Bowtie alignment analysis tool, we know that the total number of fully matched reads accounts for half of the total number of reads, which is slightly higher than the number of incompletely matched reads (Figure 1).

### 3.2. Gene Region Annotation

We found that the proportion of unique clean reads in all groups was stable at about 0.02% by calculating the number of total clean reads and unique clean reads (Table 3). Quantitative analysis matched with known miRNAs is helpful to evaluate the expression and distribution characteristics of miRNAs in samples.

Figure 2 will help us better understand the structure and composition of the genome, as well as the function and role of repetitive sequences in different biological processes.

The precursor structure, expression count, and secondary structure of the reads’ sequence of the known miRNA genes are intuitively reflected in Figure 3. This information is crucial for the study of miRNA structure and helps to reveal the mechanism of miRNA in biological processes.

In addition, the total number of clean reads completely matched with gene exons and introns was also counted, which is shown in Figure 4a. We found that this expression pattern showed a consistent trend in nine different samples, which is reflected in Figure 4b.

### 3.3. Differentially Expressed miRNA of Three Photoperiods in the Ovine Thyroid Gland

Under the three different conditions of LP42 vs. SPLP42, SP42 vs. LP42, and SP42 vs. SPLP42, 104 differentially expressed miRNAs were found. Specifically, 36 miRNAs were differentially expressed in the LP42 vs. SPLP42 group, of which 24 were down-regulated and 12 were up-regulated. In the SP42 vs. LP42 group, 39 miRNAs were differentially expressed, of which 10 were down-regulated, but the number of up-regulated miRNAs was not mentioned. In the SP42 vs. SPLP42 group, 29 miRNAs were differentially expressed, of which 13 were down-regulated and 16 were up-regulated (Figure 5). These data suggest that miRNA expression patterns vary under different conditions and may reflect different biological processes or responses.

GO annotation and KEGG pathway annotation analysis was performed on the gene sources of differentially expressed miRNAs to find the top ten GO terms and KEGG enrichment pathways with high confidence. The further enrichment of differentially expressed miRNAs was found in the KEGG pathway (*p* < 0.05). The top 10 GO terms significantly enriched by DE miRNAs belonged to three GO types: molecular function (MF), cellular components (CCs), and biological processes (BPs). DE miRNA enriched the source genes related to the cell part, binding, and cell process (Figure 6b,d,f).

## 4. lncRNA-miRNA-mRNA Co-Expression Network Construction

To explore the role and regulatory mechanism of lncRNA, miRNA, and mRNA in sheep reproduction, we used Cytoscape software to construct a network of lncRNA, miRNA, and mRNA interaction. This network helps us to understand how miRNA affects the reproductive capacity of sheep by regulating lncRNA and its target mRNA. To better understand the relationship between lncRNAs, mRNAs, and miRNAs, we used DEM and targets (DEGs) to construct an interaction network of lncRNAs-mRNAs-miRNAs to explore the molecular mechanism of different photoperiods affecting sheep thyroid transcriptome. Overall, two DE miRNAs (novel miRNAs) in SP42 vs. SPLP42 were predicted to target 42 genes. Then a lncRNA-mRNA-miRNA co-expression network was constructed, in which two DEGs were targeted by the same two new miRNAs (Figure 7a). Concerning LP42 vs. SPLP42, the 14 new predicted DE miRNAs were found to target 148 genes, and then another lncRNA-mRNA-miRNA co-expression network was constructed, in which one lncRNA and one mRNA were targeted by three new miRNAs. In addition, MSTRG.233511 is also abundantly distributed in the network (Figure 7b).

## 5. Discussion

Before our study, previous studies have successfully identified thyroid-related circRNA and lncRNA information using experimental methods [29,30]. Appropriate time and appropriate thyroid hormone secretion are of great significance for maintaining normal reproductive function [3,31]. Thyroid hormones indeed play a key role in regulating photoperiodic responses [32]. Thyroid hormones help regulate reproductive-related physiological processes, such as ovulation [33] and embryonic development [34], by affecting gene expression.

In sheep, long non-coding RNA (lncRNA) is not only present in reproductive-related tissues [26,35,36], but it also plays a vital role throughout the reproductive process [26,37,38]. The study of lncRNA, mRNA, and miRNA can complement each other and jointly promote our understanding of the regulatory network of gene expression.

In the regulation of ovarian function, miRNAs are involved in a variety of signaling pathways, affecting follicular development, ovulation, and luteal function [39]. In addition, miRNAs are being studied as biomarkers for the diagnosis and monitoring of reproductive diseases [40]. miRNA has a wide range of functions in the physiological process of preterm birth [41,42,43,44]. Some families of miRNAs can be used as uterine quiescence and contractility modulators during pregnancy or full-term or premature delivery [45].

mRNA, also known as messenger RNA, represents a class of RNA molecules and plays an important role in the regulation of gene expression [46]. In addition, the study of mRNA has important application value in the fields of gene editing [37], gene therapy [47,48], biotechnology [1], and so on. In addition, recent studies have shown that mRNA can also affect gene expression through epigenetic mechanisms [49].

The most enriched pathway between SP42 and LP42 is olfactory transduction. Among SP42 vs. SPLP42, the most significant enrichment pathway is the Wnt signaling pathway; Apelin signaling pathway; and, for miRNAs, the cancer pathway. In LP42 vs. SPLP42, the two most enriched pathways were endocrine and other factor-regulated calcium reabsorption and cocaine addiction. The enrichment pathways found in each group were not the same, but the final results can regulate the reproduction of animals by changing the photoperiod, so we can think about the role it plays in different photoperiods and the mechanisms of this role. In this study, we constructed a lncRNA-mRNA-miRNA network diagram containing three control groups to reveal the differences between them. We found that two miRNAs (Novel_369 and Novel_370) showed significant overlap in the constructed network. Differentially expressed miRNAs may be involved in the regulation of specific physiological or molecular mechanisms under these conditions. Future studies should explore the specific functions of these miRNAs and their role in regulation. These two miRNAs target five genes: gene14747, gene16852, gene17695, gene3572, and gene7823. However, the role of the target gene is not clear. Only gene14747 is a member of the glycosyl hydrolase 17 family and gene3572 is a member of the glycosyl hydrolase 8 family. Moreover, most of the differentially expressed miRNA genes are generally enriched in the Wnt signaling pathway, Apelin signaling pathway, and miRNA cancer pathway in the olfactory transduction, endocrine, and other mechanisms that regulate the calcium reabsorption pathway. Next, we focus on the analysis of these pathways: the Wnt signaling pathway mainly focuses on its relationship with the photoperiod and reproduction. In the study of Alisa Boucsein et al. [50] it can be intuitively found that this pathway is closely related to the photoperiod and circadian regulation. Others play a certain role in cancer [51], osteoporosis [52], and energy regulation [53]; the Apelin signaling pathway has a certain effect on the reproductive regulation of the hypothalamus [54], and this pathway is related to the treatment of cancer [55], glucose and lipid metabolism [56], and cardiovascular disease [57]. The calcium reabsorption pathway was found to be related to infertility [58] and prenatal stress [59], and most of the rest was related to the function of the kidney [60,61].

Studies have shown that IPMK-1 of caenorhabditis elegans can regulate the biological rhythm of animals, and inositol polyphosphate kinase IPMK-1 regulates development by regulating the calcium signaling pathway in cryptobacterium [62]. The most important calcium signaling pathway in KEGG terms is related to prolificacy [63]. Our study shows that miRNAs may play a role in thyroid development and seasonal estrus in sheep through these pathways. The miRNAs and lncRNAs found in this study are all attributed to regulatory factors. Regulatory factors regulate the physiological state of animals through various ways to reduce seasonal restrictions on animal breeding sites to a greater extent.

## 6. Conclusions

Studies have found that key miRNAs regulate the development of sheep thyroid by mediating the Wnt signaling pathway, Apelin signaling pathway, and the role of miRNAs in cancer (involving olfactory transduction, endocrine regulation, etc.), thereby affecting the activity of signaling pathways such as calcium reabsorption. In addition, competitive endogenous RNA network analysis revealed that two miRNAs (Novel_369 and Novel_370) may play a role in thyroid development and seasonal estrus in sheep through these pathways. These miRNAs may play a key role in seasonal reproduction, affecting the reproductive capacity of ewes.

## Figures and Tables

**Figure 1 animals-15-00011-f001:**
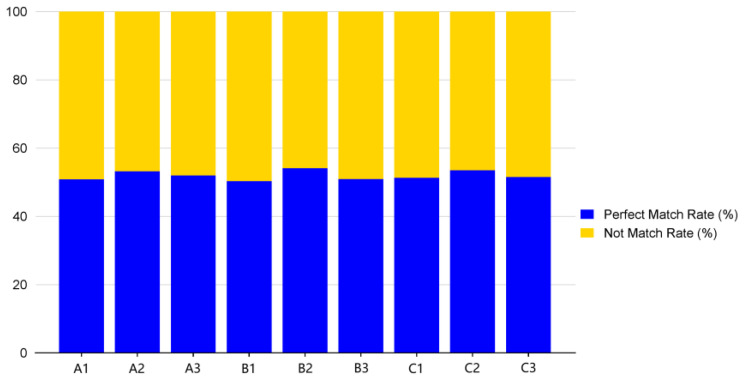
Analysis chart showing clean reads and reference sequence alignment results.

**Figure 2 animals-15-00011-f002:**
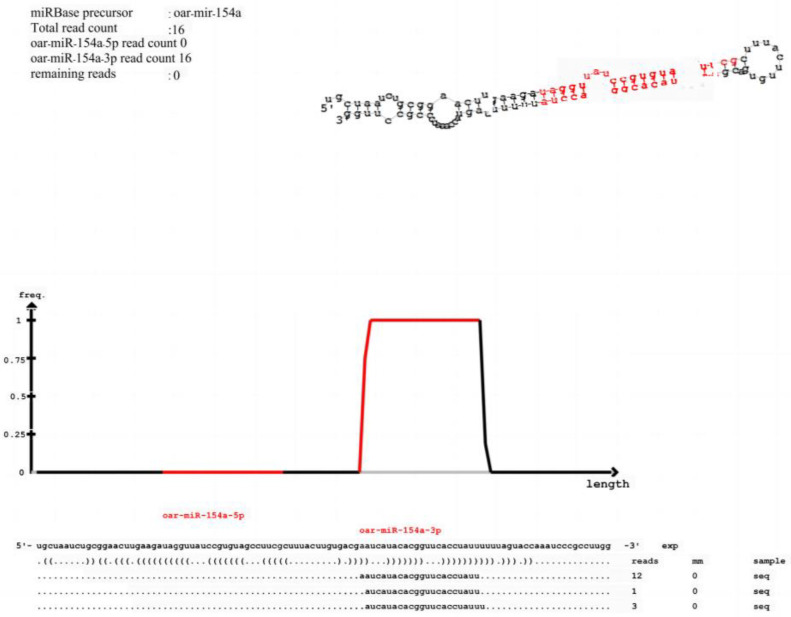
The secondary structure diagram of known miRNA genes.

**Figure 3 animals-15-00011-f003:**
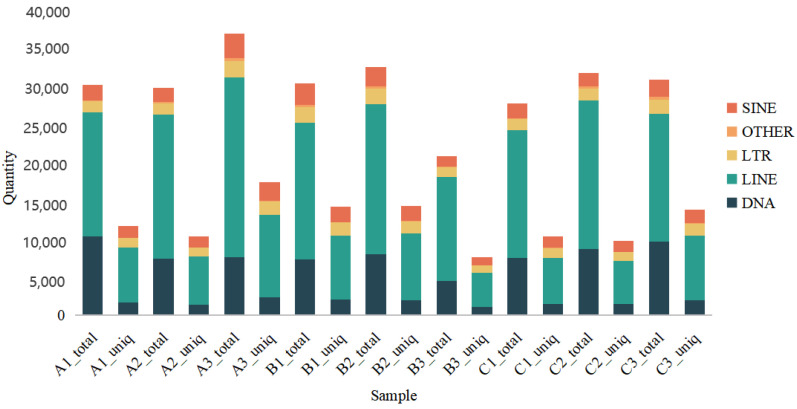
Example diagram of the secondary structure of known miRNA.

**Figure 4 animals-15-00011-f004:**
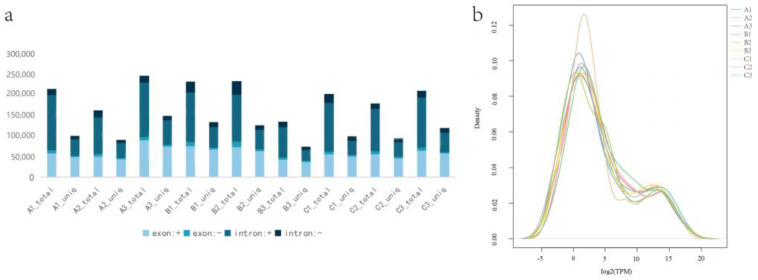
Identification of miRNA in sheep thyroid under different photoperiods. (**a**) Gene matching histogram (Exon+: the number of clean reads matching the exon sense chain; exon-: the number of clean reads matching the exon negative chain; intron+: the number of clean reads that match the intron sense chain; intron-: the number of clean reads that match the intron negative strand). (**b**) FPKM distribution of each sample.

**Figure 5 animals-15-00011-f005:**
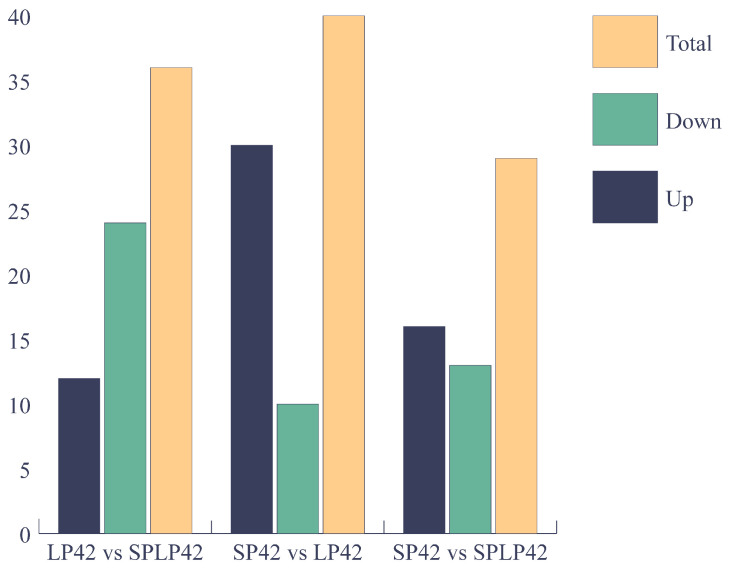
Analysis of miRNA differentially expressed in LP42 vs. SPLP42, SP42 vs. LP42, and SP42 vs. SPLP42.

**Figure 6 animals-15-00011-f006:**
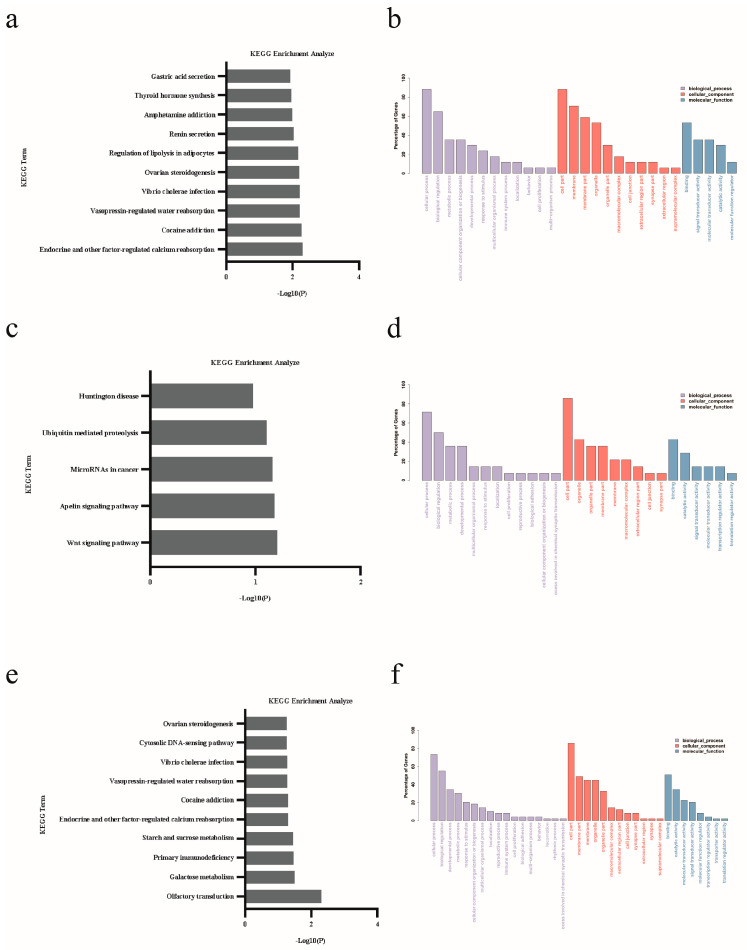
KEGG and GO function analysis (in SP42 vs. LP42, SP42 vs. SPLP42, and LP42 vs. SPLP42, respectively, 10 major miRNA KEGG enrichment pathways were observed in (**a**,**c**), and (**e**,**b**,**d**) and (**f**) show the GO function analysis of miRNA in SP42 vs. LP42, SP42 vs. SPLP42, and LP42 vs. SPLP42, respectively).

**Figure 7 animals-15-00011-f007:**
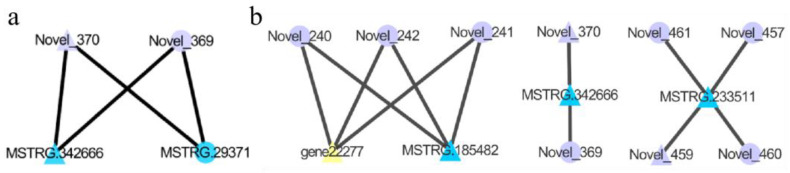
Overview of lncRNA-mRNA-miRNA networks. Purple, blue, and yellow represent miRNA, lncRNA, and mRNA, respectively. Triangle and V represent up-regulated and down-regulated, respectively. (**a**) The network diagram of SP42 vs. SPLP42 group (**b**) The network diagram of LP42 vs. SPLP42 group.

**Table 1 animals-15-00011-t001:** Test groups.

Group	Light Treatment	Number
Room 1	Short photoperiod (SP)	3
Room 2	Long photoperiod (LP)	3
Room 3	short photoperiod to a long photoperiod (SPLP)	3

Note: LP: 16 h of light (above 200 lux), 8 h of dark (within 5 lux); SP: 8 h of light, 16 h of darkness.

**Table 2 animals-15-00011-t002:** Sample information sheet.

Sample	A1	A2	A3	B1	B2	B3	C1	C2	C3
Mapping rate (%)	50.86	53.19	52.05	50.38	54.15	50.98	51.31	53.51	51.58
Known miRNA number	117	127	128	136	129	123	117	124	139
Novel miRNA number	73	76	91	91	89	65	114	82	73

**Table 3 animals-15-00011-t003:** Known miRNA.

Sample	A1	A2	A3	B1	B2	B3	C1	C2	C3
Total clean reads (mature)	8,066,453	8,940,877	9,139,311	7,944,340	11,095,413	8,491,940	8,689,677	9,725,668	7,425,343
Unique clean reads (mature)	1493	1494	1445	1627	1508	1403	1516	1387	1692

## Data Availability

The data presented in the study are deposited in the NCBI repository under the accession number PRJNA856274.

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
