# Peer review of "The Response of the miRNA Profiles of the Thyroid Gland to the Artificial Photoperiod in Ovariectomized and Estradiol-Treated Ewes"

_animals, 2024, doi:10.3390/ani15010011_

Round 1

Reviewer 1 Report

Comments and Suggestions for Authors

Dear authors,

Congratulations on your study, which, although fundamentally oriented, could represent an important element for the future development of the sheep farming industry. Methods of positively influencing photoperiodism have also been studied recently, with the goal of significantly improving meat and milk production.

Although the article contains valuable data, the main limitation is the way the information is structured and presented. The study must begin with a clear objective, outlining the specific goals to be pursued. The sections should not be mixed; the Materials and Methods section must allow for the replication of the study and should be thoroughly explained. The Results and Discussion sections, especially when dealing with two or more groups, should be organized around these groups.

A few more objective recommendations can be found below:

Title, Simple Summary, Abstract: A multitude of abbreviations are used in these parts of the article without being previously introduced, which significantly hinders easy comprehension of the text. Although many of them are established and well-known terms, they must first be introduced in their full form before being used in their abbreviated form.

Lines 24-26 Here, you need to be more specific and explain how these factors influence photoperiodism, as this is largely the main focus of the article.

Line 29 change will provide with “provided”

Line 35 complete the sentence here with “and milk”

Lines 35-37 change with this for clarity “In sheep, light stimulates the retina and transmits signals to the pineal gland through the Suprachiasmatic Nuclei (SCN), leading to changes in the duration of melatonin (MEL) secretion by the pineal gland during the night.”

There are numerous editing errors throughout the text, which need to be thoroughly revised. Most issues involve missing or incorrect spacing. For example, the correct formatting is: (MEL), not ( MEL ); ccc [23], not ccc[23]; and ccc [23,24], not ccc [23, 24].

Lines 79-83: A summary of the working method does not belong here, especially considering that this will be thoroughly developed in the Materials and Methods section. The introduction should conclude with the central aim of the article and emphasize this clearly. However, before stating the objective, a connection should be made to the broader cyclical implications that the article might have, highlighting the natural and hormonal methods used today to combat photoperiodism. For support, you might cite the article: "Comparative data about estrus induction and pregnancy rate on Lacaune ewes in non-breeding season after melatonin implants and intravaginal progestagen."

Line 95 I suggest including the body condition score (BCS) of the ewes here, as it provides important context and could influence the interpretation of the results.

Lines 95-97 A brief introduction about OVX + E2 is necessary here.

Line 100 feeding or light?

Lines 100-108 Initially, the groups were named SP and LP, but later they were referred to as SP42 and LP42. This inconsistency needs to be corrected to maintain clarity and coherence throughout the text. The naming convention should be uniform from the beginning.

It also needs to be clearly explained that the exposure was conducted over a period of 42 days. For Group 3, it would be helpful to include a graph along with an explanation of what the progression of light and dark periods entails. This will provide readers with a visual and detailed understanding of the gradual changes in photoperiod exposure applied to this group.

Lines 164-199  The first part of the results is actually a description of the methodology, and it should be moved to the Materials and Methods section. This will ensure that the structure of the article is consistent and that the results section focuses strictly on the findings and data analysis.

Lines 262-352: The Discussion section needs to focus on the results of the study. Currently, these paragraphs serve more as an introduction to the topic rather than an analysis of the findings. The discussion should interpret the results, compare them with previous studies, and explore their implications, rather than providing background information or general context on the subject. From such an article, it would also be expected that the Discussion section includes a comparison between the three groups, highlighting the significant differences. This comparison should analyze how the results differ across the groups and what those differences imply in the context of the study, drawing attention to any meaningful trends or findings.

Author Response

Dear editor and reviewer:

Thank you for your letter and the reviewers' comments concerning our manuscript entitled “Response of the miRNA Profiles of the Thyroid Gland to Artificial Photoperiod in OVX+E2 Ewes”.Those comments are all valuable and very helpful for revising and improving our paper, as well as the important guiding significance to our research. We have studied comments carefully and have made corrections which we hope meet with approval.

Comments 1: Title, Simple Summary, Abstract: A multitude of abbreviations are used in these parts of the article without being previously introduced, which significantly hinders easy comprehension of the text. Although many of them are established and well-known terms, they must first be introduced in their full form before being used in their abbreviated form.

Response 1: Thank you very much for pointing out this point, this is my negligence, I have in the text where the need for correction, due to the order of change being a bit confused, please forgive me here can not give you a clear indication of the position in the text

Comments 2: Lines 24-26 Here, you need to be more specific and explain how these factors influence photoperiodism, as this is largely the main focus of the article.

Response 2: Thank you very much for putting forward this suggestion. I have perfected it in this paper. You can see it in L26-L28.

Comments 3: Line 29 change will provide with “provided”.

Response 3: Thank you very much for putting forward this suggestion. I have corrected.(L30)

Comments 4: Line 35 complete the sentence here with “and milk”

Response 4: Thank you for your correction, I have corrected the text. (L39)

Comments 5: Lines 35-37 change with this for clarity “In sheep, light stimulates the retina and transmits signals to the pineal gland through the Suprachiasmatic Nuclei (SCN), leading to changes in the duration of melatonin (MEL) secretion by the pineal gland during the night.”

Response 5: Thank you for your correction, I have corrected the text. (L39-41)

Comments 6: There are numerous editing errors throughout the text, which need to be thoroughly revised. Most issues involve missing or incorrect spacing. For example, the correct formatting is: (MEL), not ( MEL ); ccc [23], not ccc[23]; and ccc [23,24], not ccc [23, 24].

Response 6: Thank you very much for pointing out this valuable advice, clear and clear solution to my doubts about such issues, has been corrected in the text, due to too much, can not be marked one by one, please forgive me

Comments 7: Lines 79-83: A summary of the working method does not belong here, especially considering that this will be thoroughly developed in the Materials and Methods section. The introduction should conclude with the central aim of the article and emphasize this clearly. However, before stating the objective, a connection should be made to the broader cyclical implications that the article might have, highlighting the natural and hormonal methods used today to combat photoperiodism. For support, you might cite the article: "Comparative data about estrus induction and pregnancy rate on Lacaune ewes in non-breeding season after melatonin implants and intravaginal progestagen."

Response 7: Thank you very much for your suggestion, I have put this passage here in the working methods it should be in. Then you recommend this article, again thank you for your busy schedule in providing me with an interesting reference article, but I take into account the coherence of the article, temporarily do not know where to put the right, so temporarily put this aside, I hope you do not mind (L108-111).

Comments 8: Line 95 I suggest including the body condition score (BCS) of the ewes here, as it provides important context and could influence the interpretation of the results.

Response 8: Received your advice and thank you, but the ewe did not record its body condition score at that time, but we ensured that they were healthy multiparous ewes. (90)

Comments 9: Lines 95-97 A brief introduction about OVX + E2 is necessary here.

Response 9: I agree(L87-89).

Comments 10: Line 100 feeding or light?

Response 10: Thank you for your suggestion, yes, the ewe is an adult multiparous ewe, after feeding free diet, and then according to the light time different points of the room.

Comments 11: Lines 100-108 Initially, the groups were named SP and LP, but later they were referred to as SP42 and LP42. This inconsistency needs to be corrected to maintain clarity and coherence throughout the text. The naming convention should be uniform from the beginning.

Response 11: Thank you for your correction, I have corrected it.

Comments 12: It also needs to be clearly explained that the exposure was conducted over a period of 42 days. For Group 3, it would be helpful to include a graph along with an explanation of what the progression of light and dark periods entails. This will provide readers with a visual and detailed understanding of the gradual changes in photoperiod exposure applied to this group.

Response 12: I agree it. I have added Table 1 for reference. (L98-100)

Comments 13: Lines 164-199 The first part of the results is actually a description of the methodology, and it should be moved to the Materials and Methods section. This will ensure that the structure of the article is consistent and that the results section focuses strictly on the findings and data analysis.

Response 13: Agree. I

Comments 14: Lines 262-352: The Discussion section needs to focus on the results of the study. Currently, these paragraphs serve more as an introduction to the topic rather than an analysis of the findings. The discussion should interpret the results, compare them with previous studies, and explore their implications, rather than providing background information or general context on the subject. From such an article, it would also be expected that the Discussion section includes a comparison between the three groups, highlighting the significant differences. This comparison should analyze how the results differ across the groups and what those differences imply in the context of the study, drawing attention to any meaningful trends or findings.

Response 14: Thank you for your suggestion, it has been corrected(L251-278)

Thank you very much for your attention and time. Look forward to hearing from you.

Yours sincerely.

Reviewer 2 Report

Comments and Suggestions for Authors

This study attempts to explain how Response of the miRNA Profiles of the Thyroid Gland to Artificial Photoperiod in OVX+E2 Ewes.” .

I listed my other concerns below in the order I found them in the manuscript.

Simple Summary

Well-written.

Abstract

Well-written.

-        L23-24. How were the samples collected? When were the samples collected? How many samples were collected?

-        L29-30. The conclusion is not related to the objectives of the experiment (L17-19)

Introduction

The authors must add evidence in the Introduction that the Sunite breed is seasonal and the variation in photoperiod in the study site.

-        L33-34. Not all the breeds. Please be more specific.

-        L55-61. I do not see the relevance of these sentences to the experiment.

-        L79-84. These data should be presented in M&M.

Material and Methods

-        L95. What about age? Were the ewes of similar age?

-        L96-97. Raised? From birth to the start of the experiment?

-        L98-99. More details about the diet. Does it meet the nutritional requirements for maintenance? Gain? Was the diet provided ad libitum?

-        L100-105. Please provide the LUX intensity.

-        L106-108. More details about this procedure are needed. The reader should be able to replicate the experiment. How were the samples collected?

-         

Results

The results should be concise.

-        L154-205. This methodology should be presented in the M&M section.

-        L213-218. This section could be moved to the Discussion section.

-        L222-237. The results should be concise. In this paragraph, the authors are discussing the results.

Discussion

The discussion needs to be improved.

-        L300-350. Although the information is interesting, the authors do not discuss the results. This information could have been presented in the Introduction.

Author Response

Dear editor and reviewer:

Thank you for your letter and for the reviewers' comments concerning our manuscript entitled “Response of the miRNA Profiles of the Thyroid Gland to Arti-ficial Photoperiod in OVX+E2 Ewes”.Those comments are all valuable and very helpful for revising and improving ourpaper, as well as the important guiding significance to our researches. We have studied comments carefully and have made correction which we hope meet with approval.

Comments 1: Abstract

L23-24. How were the samples collected? When were the samples collected? How many samples were collected?

L29-30. The conclusion is not related to the objectives of the experiment (L17-19)

Response 1:

Thank you very much for your suggestions, about how to collect samples and collection methods I have added in the materials and methods, in the summary also added some description accordingly(L23-24→L19-21);

I have deepened the explanation of the relationship between the conclusion and the result. I hope you can give valuable opinions (L29-30→L27-28)

Comments 2: Introduction

The authors must add evidence in the Introduction that the Sunite breed is seasonal and the variation in photoperiod in the study site.

L33-34. Not all the breeds. Please be more specific.

L55-61. I do not see the relevance of these sentences to the experiment.

L79-84. These data should be presented in M&M.

Response 2:

For Sunite sheep is a typical seasonal sheep I have added a description(L38-39);

Thank you for pointing out these views, I agree with you, and in the text I made some amendments(L33-34→L38-39);

I agree with your opinion, I have deleted these words. (L55-61);

After careful reading, I put these in the M&M part. (L79-84→L110-115)

Comments 3: Material and Methods

L95. What about age? Were the ewes of similar age?

L96-97. Raised? From birth to the start of the experiment?

L98-99. More details about the diet. Does it meet the nutritional requirements for maintenance? Gain? Was the diet provided ad libitum?

L100-105. Please provide the LUX intensity.

L106-108. More details about this procedure are needed. The reader should be able to replicate the experiment. How were the samples collected?

Response 3:

Regarding the collection of samples, I have made specific modifications in the text. (L89-102)

Comments 4: Results

L154-205. This methodology should be presented in the M&M section.

L213-218. This section could be moved to the Discussion section.

L222-237. The results should be concise. In this paragraph, the authors are discussing the results.

Response 4:

Thank you for your suggestions. I agree with them. According to your opinion, I modified these in the text, after the arrow is corrected in the text after the location.

(L154-205→Dispersed display in the M&M part)

(L213-218→L263-266)

(L222-237→L253-257)

Comments 5: Discussion

L300-350. Although the information is interesting, the authors do not discuss the results. This information could have been presented in the Introduction.

Response 5: Thank you very much for your suggestion. I have modified it. (L300-350→L231-252)

Thank you very much for your attention and time. Look forward to hearing from you.

Yours sincerely

Round 2

Reviewer 1 Report

Comments and Suggestions for Authors

Dear authors, the article looks better after the revisions and is taking a form suitable for publication. However, there are still terms, especially in the first part of the article and even in the title, that appear in abbreviated form the first time and are only later described. Additionally, some suggestions would be welcome for framing the study within the clinical field by highlighting directions for the applicability of influencing seasonality in this species. A complete revision of the entire text is also necessary, as there are still editing errors that have not yet been corrected.

Author Response

Comments 1: However, there are still terms, especially in the first part of the article and even in the title, that appear in abbreviated form the first time and are only later described.

Response 1: Thank you very much for your advice, I carefully modified this error in the text, as you say the title of the abbreviation, I will write the first full name in the summary, highlighted. There are other abbreviations I have highlighted in the text display. (L3, L47, L20)

Comments 2: Additionally, some suggestions would be welcome for framing the study within the clinical field by highlighting directions for the applicability of influencing seasonality in this species.

Response 2: Thank you very much for your advice, I have summarized it in the text. (L419-422)

Comments 3: A complete revision of the entire text is also necessary, as there are still editing errors that have not yet been corrected.

Response 3: Thank you very much for your advice, I have carefully read the full text, which has been modified after several errors, and highlighted the display. (L31, L56, L72, L394, L403-405, L429)

Reviewer 2 Report

Comments and Suggestions for Authors

The manuscript has been improved.

Author Response

Thank you very much for your revision of the article.